# Depth-Dependent Control in Vision-Sensor Space for Reconfigurable Parallel Manipulators

**DOI:** 10.3390/s23167039

**Published:** 2023-08-09

**Authors:** Arturo Franco-López, Mauro Maya, Alejandro González, Antonio Cardenas, Davide Piovesan

**Affiliations:** 1Facultad de Ingenieria, Universidad Autonoma de San Luis Potosi, San Luis Potosi 78290, Mexico; arturo.franco@uaslp.mx (A.F.-L.); antonio.cardenas@uaslp.mx (A.C.); 2Escuela de Ingeniería y Ciencias, Tecnologico de Monterrey, Querétaro 76130, Mexico; a.gonzalezda@tec.mx; 3Biomedical, Industrial and Systems Engineering Department, Gannon University, Erie, PA 16541, USA; piovesan001@gannon.edu

**Keywords:** vision-based control, camera-space manipulation, parallel robot

## Abstract

In this paper, a control approach for reconfigurable parallel robots is designed. Based on it, controls in the vision-sensor, 3D and joint spaces are designed and implemented in target tracking tasks in a novel reconfigurable delta-type parallel robot. No a priori information about the target trajectory is required. Robot reconfiguration can be used to overcome some of the limitations of parallel robots like small relative workspace or multiple singularities, at the cost of increasing the complexity of the manipulator, making its control design even more challenging. No general control methodology exists for reconfigurable parallel robots. Tracking objects with unknown trajectories is a challenging task required in many applications. Sensor-based robot control has been actively used for this type of task. However, it cannot be straightforwardly extended to reconfigurable parallel manipulators. The developed vision-sensor space control is inspired by, and can be seen as an extension of, the Velocity Linear Camera Model–Camera Space Manipulation (VLCM-CSM) methodology. Several experiments were carried out on a reconfigurable delta-type parallel robot. An average positioning error of 0.6 mm was obtained for static objectives. Tracking errors of 2.5 mm, 3.9 mm and 11.5 mm were obtained for targets moving along a linear trajectory at speeds of 6.5, 9.3 and 12.7 cm/s, respectively. The control cycle time was 16 ms. These results validate the proposed approach and improve upon previous works for non-reconfigurable robots.

## 1. Introduction

Parallel robots have received increased attention in recent years, and one of the most popular applications at an industrial level is pick and place based on the parallel delta robot [1]. A parallel robot can be described as a manipulator composed of a fixed and a mobile platform connected by two or more kinematic chains generating closed loops with the end effector mounted on the mobile platform [2,3]. The payload is therefore distributed among the kinematic chains; besides, the actuators are mounted directly in the fixed base, which means that the kinematic chains do not carry them. This results in thinner and lighter links. Parallel robots have some advantages over serial robots: higher positioning accuracy, better rigidity, greater load-capacity-to-weight ratio, higher speeds and accelerations, lower inertia, good stability, and simple inverse kinematics [3,4,5,6]. On the other hand, parallel robots have some disadvantages: relatively small and complex workspace, often having complex singularities inside the workspace: complex geometric (position relationships), kinematic (velocity relationships) and dynamic models that complicate their modeling and control design [5,7,8].

### 1.1. Reconfiguration and Redundancy on Parallel Robots

Reconfiguration has the ability to facilitate the adaptation and optimization of parallel robots to a wide range of applications. It overcomes some of their typical difficulties such as the reduced workspace. Reconfiguration on parallel robots may involve changes to the length of the arms, the orientation of the kinematic chains, or the position of the mobile platform in relation to the fixed base. These changes can be made manually or automatically using intelligent control systems. The implementation of a reconfigurable robot in which the reconfiguration is performed via a mechanism that can be controlled in operation time makes the robot redundant [9,10], which requires an even more complex control design.

In parallel robots, there are multiple types of redundancy, the most important being the kinematic redundancy and the actuator redundancy. A way to generate kinematic redundancy consists of adding active joints to the kinematic chains [11,12,13,14]. This implies that there is an infinite combination of joint velocities producing the same specific velocity of the end effector. Actuator redundancy consists of having more than the necessary kinematic chains between the fixed and mobile platforms. From the static point of view, for each force vector and torque at the end effector, there will be an infinite number of force and wrench vectors for the active joints [15]. For redundant robots, obtaining a single solution to the inverse kinematic problem is called solving the redundancy problem.

Redundancy has great advantages in parallel robots, such as enhancing the workspace, avoidance or reduction of singularities, increasing the payload capacity, improving the dynamics of the mechanism, and eliminating motion clearance, among others [5,16,17]. However, these advantages come at a price such as increased complexity in the kinematic and dynamic models, challenging control design, and higher energy consumption due to extra actuators and internal forces/torques [18]. Moreover, extra actuators imply extra costs and introduce additional friction from the supplementary active joints. In summary, robot reconfiguration can be used to tackle some of the problems of parallel robots at the cost of increasing the complexity of the manipulator, resulting in a more challenging control design.

### 1.2. Camera Space Manipulation

Providing parallel robots with sensors that enable them to monitor the environment around them is another effective method of extending the range of applications for these machines. With this information, the robot is able to adapt to its environment and the events occurring in it. In particular, for an object-tracking task, it is important for a robot to be able to respond appropriately to changes in the environment. Vision systems are one of the most frequently used options to obtain information about the interaction of the robot with the environment due to their ease of operation [19]. Vision-based control is a sensor-based control strategy, strongly resembling the process of our central nervous system [20]. Among the control schemes that use vision systems, two of the most general and popular are Visual Servoing (VS) [21,22,23] and Camera Space Manipulation (CSM) [24,25]. VS control is commonly applied for real-time control, incorporating visual information from one or more cameras in a closed-loop control cycle. It provides increased precision in manipulation tasks based on visual feedback [22,26,27]; it also requires the Image Jacobian, which depends on the visual features used and the camera parameters (obtained through a calibration process) and can be complex and difficult to obtain [23]. In its first applications, the CSM methodology [24] and its variants, such as Linear Camera Model–Camera Space Manipulation (LCM-CSM) [28] consisted of an open-loop position level control. One of the most interesting features is that it is not necessary to know the view parameters of the camera a priori. Instead, a data capture is performed in which either a set of points in three-dimensional coordinates or a set of joint coordinates are mapped to their corresponding coordinates in image space to obtain such parameters. A practical way to accomplish this is by placing visual markers on the robot’s end effector and using the geometric model of the camera to associate the coordinates of the markers with their corresponding coordinates in camera space. This circumvents the need to know a priori the Jacobian transformation of the manipulator, which can be a complex task [29,30]. With the data set and the camera model, the estimation of the so-called vision parameters is performed. This process is called “preplan”. An even simpler method is to approximate the orthographic camera model with the pin-hole camera model obtaining the LCM-CSM [28] variant where the view parameters are estimated from a linear model. Recent work has derived a camera–Jacobian matrix from calculating the derivative with respect to time for a pin-hole camera model. This allowed the development of the VLCM-CSM variant enabling the design of velocity-based closed-loop control laws [6].

Some vision techniques have been reported in the literature to guide parallel robots through various tasks. In [29], a vision system is used to estimate the pose of the Hexa parallel robot, while [31] uses a computer vision system to develop an obstacle collision detection method for the same robot, no control description is included in either work. In [32], computer vision is used to identify the geometric parameters of a Delta parallel architecture. Regular PIDs are then used to control the robot joints. Ref. [33] uses computer vision and an unscented Kalman filter to estimate the pose of a planar 3RRR parallel robot and uses a PID control to guide the robot through a point to point path-tracking task. In [34,35], VS is used to control a cable-driven parallel robot, pose-based (3D and 2D) schemes are developed as well as a mix of them (2D 1/2); a conservative sufficient condition for stability is obtained. Ref. [6] develops the VLCM-CSM approach to control a Delta-type parallel robot in target tracking tasks. In summary, there are still few works developing vision-based control for parallel robots, some of them are used only to detect the end-effector pose to then guide it to complete the task. However, to the best of the authors’ knowledge, no general control design approach for reconfigurable parallel robots has been reported in the literature.

This work presents a control approach for reconfigurable parallel robots. Following this approach, controls are designed and implemented in a novel reconfigurable Delta-type parallel robot [16]. In this work. the geometric and kinematic models are derived for this robot for the first time, yielding an initial contribution of this paper. Moreover, the control design in image space is based on the VLCM-CSM approach. For this, an original derivation of the image Jacobian associated with this approach is obtained, where it is shown that the only variable is a factor depending on the depth of the observed feature. This makes this Jacobian extremely simple to implement and update on each control cycle. This simple Jacobian can be used for any variant of the VLCM-CSM methodology and constitutes a second contribution of this work. Using the aforementioned elements, a general control approach, and main contribution of this work, for reconfigurable parallel robots was developed. Based on this approach, control laws in the the vision-sensor, 3D and joint spaces are designed and implemented. The controls in the 3D and joint spaces have the limitation that it is in general difficult to obtain the reference information (target position) in such spaces, particularly in the case of a target moving along an unknown trajectory. Usually this information has to be obtained through indirect estimates using available measurements and position reconstruction models prone to errors. This limitation is intrinsic to any control design in these spaces. The control in the vision sensor space does not require reconstruction as this type of sensor provides direct measurements of the target in that space regardless of its trajectory, this results in a robust approach.

This article is organized as follows: Section 2 analyzes the kinematic model of the reconfigurable delta robot considered in this work. Section 3 presents an original derivation of the image Jacobian based on the VLCM-CSM methodology. In Section 4, a control approach for reconfigurable parallel robots is developed and several control laws based on this approach are synthesized for a reconfigurable delta-type parallel robot. Section 5 details the experimental setup and the experiments performed. Section 6 presents and discusses the results. Finally, Section 7 describes the authors’ conclusions of this work and outlines opportunities for future work.

## 2. Modeling of the Reconfigurable Delta Robot

### 2.1. Description of the Reconfigurable Delta Robot

Let us consider the delta robot in Figure 1; it consists of two platforms, one fixed and a mobile one, connected by three kinematic chains (numbered with index i=1,2,3), evenly distributed about the vertical axis *Z* of the robot’s fixed frame. In this typical configuration, the fixed platform is on top of the mechanism (horizontal element), and the mobile platform (i.e., the part where the end-effector is attached) is a small, horizontal element at the bottom of the mechanism. Each chain has two links; one end of the chain is connected to the fixed platform through an actuated, rotational joint placed at point $A_{i}$ with joint variable $\theta_{i1}$, and the other end is connected to the mobile platform through a passive universal joint (some designs use a spherical one), at point $C_{i}$. The two links on each chain are connected together at point $B_{i}$ through a passive universal (can be spherical) joint with (passive) joint variables $\theta_{i2}$ and $\theta_{i3}$. The use of links with a parallelogram configuration, together with the robot architecture, ensures the parallelism between both platforms at all times. *R* is the radius of the fixed platform (i.e., the distance between the origin of the robot’s frame and point $A_{i}$), *r* is the radius of the moving platform, (i.e., the distance between the center of the platform and point $C_{i}$), $\alpha_{i}$ is the angle between axis *X* of the fixed frame and each of the three kinematic chains that make up the robot, measured about the fixed *Z* axis; $L_{1}$ and $L_{2}$ are the length of the first and second links of the kinematic chains, respectively, and *P* is the 3D end-effector position expressed in the fixed frame. With the reconfiguration capability considered in this work, *R* becomes a (joint) variable. In Figure 2, the reconfiguration mechanism for the robot is shown. More details on the reconfiguration analysis and mechanism can be found in [16].

The following key points can be defined with respect to the robot’s fixed reference frame:(1)$$A_{i}=\left[\begin{array}{c} R\cos(\alpha_{i})\\ R\sin(\alpha_{i})\\ 0 \end{array}\right]$$
(2)$$B_{i}=\left[\begin{array}{c} \left(R+L_{1}\cos\left(\theta_{i1}\right)\right)\cos\left(\alpha_{i}\right)\\ \left(R+L_{1}\cos\left(\theta_{i1}\right)\right)\sin\left(\alpha_{i}\right)\\ -L_{1}\sin\left(\theta_{i1}\right) \end{array}\right]$$
(3)$$C_{i}=\left[\begin{array}{c} P_{x}+r\cos\left(\alpha_{i}\right)\\ P_{y}+r\sin\left(\alpha_{i}\right)\\ P_{z} \end{array}\right]$$
(4)$$P=\left[\begin{array}{c} P_{x}\\ P_{y}\\ P_{z} \end{array}\right]$$

### 2.2. Forward Geometric Model

Based on the work reported in [16], and considering the reconfiguration, we define the active joints and Cartesian coordinate vectors of the end-effector as
(5)$$q=\left[\begin{array}{c} R\\ \theta_{11}\\ \theta_{21}\\ \theta_{31} \end{array}\right]$$
and
(6)$$x=P=\left[\begin{array}{c} P_{x}\\ P_{y}\\ P_{z} \end{array}\right]$$
respectively. Note that R is included in the vector of joint variables because the reconfiguration mechanism can be controlled together with the three kinematic chains. On this basis, the reconfigurable robot is then modeled as a redundant robot with four control inputs.

A common way to solve the direct kinematics problem for the delta robot is to solve the intersection of three spheres centered at points ${\tilde{B}}_{i}$ defined as
(7)$${\tilde{B}}_{i}=\left[\begin{array}{c} \begin{array}{c} B_{ix} \end{array}-r\cos\left(\alpha_{i}\right)\\ \begin{array}{c} B_{iy} \end{array}-r\sin\left(\alpha_{i}\right)\\ \begin{array}{c} B_{iz} \end{array} \end{array}\right]$$
and radii $L_{2}$. The spheres are associated with the free motion of points $C_{i}$ with respect to points $B_{i}$, and obviously do not intersect at *P*, however, by horizontally shifting the spheres (their centers) a distance *r* towards fixed axis *Z*, the shifted spheres intersect at *P*. With this, we define a sphere per each kinematic chain with center ${\tilde{B}}_{i}$, and the intersection of the three spheres with the smallest value on the *Z* axis will be P=x, which is the solution to the direct kinematics problem. This can be represented by the following system of equations: (8)$${\left(\begin{array}{c} {\tilde{B}}_{ix} \end{array}-P_{x}\right)}^{2}+{\left(\begin{array}{c} {\tilde{B}}_{iy} \end{array}-P_{y}\right)}^{2}+{\left(\begin{array}{c} {\tilde{B}}_{iz} \end{array}-P_{z}\right)}^{2}=L_{2}^{2}$$

It may be observed that this approach is valid regardless of whether R is constant or variable. For that, this solution is valid for a reconfigurable delta robot.

### 2.3. Forward Kinematic Model

Based on Figure 1, the following closed-loop geometric constraint system can be defined:(9)$$\vec{\mathrm{OP}}+\vec{{\mathrm{PC}}_{i}}=\vec{{\mathrm{OA}}_{i}}+\vec{A_{i}B_{i}}+\vec{B_{i}C_{i}}$$
which has the following matrix form:(10)$$\left[\begin{array}{c} P_{x}\cos\left(\alpha_{i}\right)-P_{y}\sin\left(\alpha_{i}\right)\\ P_{x}\sin\left(\alpha_{i}\right)+P_{y}\cos\left(\alpha_{i}\right)\\ P_{z} \end{array}\right]+\left[\begin{array}{c} r\\ 0\\ 0 \end{array}\right]=\left[\begin{array}{c} R\\ 0\\ 0 \end{array}\right]+L_{1}\left[\begin{array}{c} \cos(\theta_{1i})\\ 0\\ -\sin(\theta_{1i}) \end{array}\right]+L_{2}\left[\begin{array}{c} \sin\left(\theta_{i3}\right)\cos\left(\theta_{i2}+\theta_{i1}\right)\\ \cos(\theta_{i3})\\ -\sin\left(\theta_{i3}\right)\cos\left(\theta_{i3}+\theta_{i1}\right) \end{array}\right]$$

This can be rewritten as
(11)$$\left(x_{i}+r\right)=R+L_{1i}+L_{2i}$$

Deriving with respect to time, we obtain the following result:(12)$${\dot{x}}_{i}=\dot{R}+{\dot{L}}_{1i}+{\dot{L}}_{2i}$$
for the reconfigurable robot, *R* has a variable derivative, while in the case of the robot without reconfiguration, the derivative of *R* is zero.

Notice that the links can only move around a pivot; hence, we can simplify the previous equation in the following form:(13)$$x_{i}=R+w_{L_{1i}}\times L_{1i}+w_{L_{2i}}\times L_{2i}$$

To further simplify the equation above it is convenient to express the equation above on a reference frame located on $B_{i}$ and obtain an expression that depends only on the articular variables. To accomplish this, we multiply the above equation by the director vector $\hat{L_{2i}}$, obtaining
(14)$$\hat{L_{2i}}\cdot {\dot{x}}_{i}=\hat{L_{2i}}\cdot \dot{R}+\hat{L_{2i}}\cdot \left(w_{L_{1i}}\times L_{1i}\right)$$

Substituting Equation (Equation 10) in the left side of the above equation, we have:(15)$$\begin{array}{c} \hat{L_{2i}}\cdot {\dot{x}}_{i}=\left[\sin\theta_{i3}\cos\left(\theta_{i2}+\theta_{i1}\right)\right]\left[{\dot{P}}_{x}\cos\alpha_{i}-{\dot{P}}_{y}\sin\alpha_{i}\right]+\cos\theta_{i3}\left[{\dot{P}}_{x}\sin\alpha_{i}+{\dot{P}}_{y}\cos\alpha_{i}\right]\\ -\left[\sin\theta_{i3}\sin\left(\theta_{i2}+\theta_{i1}\right)\right]{\dot{P}}_{z}=a_{ix}{\dot{P}}_{x}+a_{iy}{\dot{P}}_{z}+a_{iz}{\dot{P}}_{z} \end{array}$$
where
(16)$$\begin{array}{c} a_{ix}=\sin\theta_{i3}\cos\left(\theta_{i2}+\theta_{i1}\right)\cos\alpha_{i}+\cos\theta_{i3}\sin\alpha_{i}\\ a_{iy}=-\sin\theta_{i3}\cos\left(\theta_{i2}+\theta_{i1}\right)\sin\alpha_{i}+\cos\theta_{i3}\cos\phi_{i}\\ a_{iz}=-\sin\theta_{i3}\sin\left(\theta_{i2}+\theta_{i1}\right) \end{array}$$

Substituting the right side of Equation (Equation 14), we have
$$\hat{L_{2i}}\cdot \left(w_{L_{1i}}\times L_{1i}\right)=-L_{1}\sin\theta_{i2}\sin\theta_{i3}{\dot{\theta }}_{i1}$$
and
$$\hat{L_{2i}}\cdot \dot{R}=\dot{R}\sin\theta_{i3}\cos\left(\theta_{i2}+\theta_{i1}\right)$$
in such a way that
(17)$$\hat{L_{2i}}\cdot \dot{R}+\hat{L_{2i}}\cdot \left(w_{L_{1i}}\times L_{1i}\right)=\sin\theta_{i3}\cos\left(\theta_{i2}+\theta_{i1}\right)\dot{R}-L_{1}\sin\theta_{i2}\sin\theta_{i3}{\dot{\theta }}_{i1}$$

Substituting in the Equations (Equation 16) and (Equation 17) for each *i* we have
$$a_{1x}{\dot{P}}_{x}+a_{1y}{\dot{P}}_{y}+a_{1z}{\dot{P}}_{z}=\sin\left(\theta_{13}\right)\cos\left(\theta_{12}+\theta_{11}\right)\dot{R}-L_{1}\sin\left(\theta_{12}\right)\sin\left(\theta_{13}\right)\dot{\theta_{11}}$$
$$a_{2x}{\dot{P}}_{x}+a_{2y}{\dot{P}}_{y}+a_{2z}{\dot{P}}_{z}=\sin\left(\theta_{23}\right)\cos\left(\theta_{22}+\theta_{21}\right)\dot{R}-L_{1}\sin\left(\theta_{22}\right)\sin\left(\theta_{23}\right)\dot{\theta_{21}}$$
$$a_{3x}{\dot{P}}_{x}+a_{3y}{\dot{P}}_{y}+a_{3z}{\dot{P}}_{z}=\sin\left(\theta_{33}\right)\cos\left(\theta_{32}+\theta_{31}\right)\dot{R}-L_{1}\sin\left(\theta_{32}\right)\sin\left(\theta_{33}\right)\dot{\theta_{31}}$$
which can be rewritten in matrix form as:(18)$$J_{x}\dot{x}=J_{q}\dot{q}$$
where
(19)$$J_{x}=\left[\begin{array}{ccc} a_{11} & a_{12} & a_{13}\\ a_{21} & a_{22} & a_{23}\\ a_{31} & a_{32} & a_{33} \end{array}\right]$$
is a Jacobian matrix that associate each point $B_{i}$ with the work-space, and where
(20)$$J_{q}=\left[\begin{array}{cccc} b_{11} & b_{12} & 0 & 0\\ b_{21} & 0 & b_{23} & 0\\ b_{31} & 0 & 0 & b_{34} \end{array}\right]$$
with
$$b_{i1}=\sin\left(\theta_{i3}\right)\cos\left(\theta_{i2}+\theta_{i1}\right)$$
$$b_{i,i+1}=-L_{1}\sin\left(\theta_{i2}\right)\sin\left(\theta_{i3}\right)$$
is a Jacobian matrix that associate the joint-space with each $B_{i}$.

When the robot is not reconfigured, these Jacobians become square and measure 3×3. The new joint variable *R* is reflected in the Jacobian $J_{q}$ and generates a non-square matrix of dimension 4×3 while the matrix $J_{x}$ is not affected and remain of dimension 3×3.

The following forward kinematic model is obtained from Equation (Equation 18):(21)$$\dot{x}=J\dot{q}$$
where $J=J_{x}^{-1}J_{q}$ is the Jacobian matrix of the robot.

The solution to the inverse kinematics problem is the solution to the redundancy problem.

### 2.4. Resolution of the Redundancy Problem

In the case of kinematic redundancy in parallel robots, the inverse kinematics problem has an infinite number of solutions. Solving the redundancy problem consists of obtaining a single solution to the inverse kinematics. One of the most common ways to obtain the inverse kinematic model is to use the following solution [12]:(22)$$\dot{q}=J^{†}\dot{x}-\gamma \left(I-J^{†}J\right)\frac{\delta \mu }{\delta q}$$
where $J^{†}$ is the Moore Penrose pseudo-inverse of J, μ is a performance index to be minimized in order to perform a second task, and the matrix $I-J^{†}J$ maps to the null space of J. The parameter γ is a scaling factor that changes the priority of the second task relative to the first. An example of such a secondary task is the minimizing of the following cost function:(23)$$\mu =\frac{1}{2}\sum_{i=1}^{n}{\left(\frac{q_{i}-q_{i}^{mid}}{q_{i}^{max}-q_{i}^{min}}\right)}^{2}$$
where $q_{i}^{mid}$ is the mid value that the *i*-th active joint can reach between $q_{i}^{max}$ and $q_{i}^{min}$, which are the maximum and minimum values that the i−th active joint can reach.

This cost function allows us to keep the active joints apart from their joint limit and thus ensure safe operations. Yet, there may be positions reachable by the unconstrained robot, but not by the robot with the actual joint limits. The gradient of (Equation 23) is given by
(24)$$\frac{\delta \mu }{\delta q}=\frac{q_{i}-q_{i}^{mid}}{{\left(q_{i}^{max}-q_{i}^{min}\right)}^{2}}$$

This solution for inverse kinematics, including the aforementioned secondary task, is used in the control laws to be defined.

### 2.5. Inverse Geometric Model

Two algorithms are proposed to solve the inverse geometric model to obtain a single solution. The first algorithm is based on the resolution of the inverse geometric model of the original delta robot, as it can be found in [16]. This algorithm consists of the following steps:(i)Solve the inverse kinematics for a specified value of R as if it were the original delta robot.(ii)Calculate the Jacobian matrix of the reconfigurable delta robot.(iii)Calculate the condition number and compare it with the smallest previous condition number of the Jacobian matrix. If the current one is smaller, the actual joint values are the solution.(iv)Repeat the process iteratively varying R over its entire range of motion, i.e., from 85 mm to 500 mm.(v)The solution with the smallest condition number is obtained at the end of all iterations.

The advantages of this algorithm are that it is easy to implement and guarantees that the smallest minimum condition number is found. As a disadvantage, it has a high computational cost and may not be suitable for applications in real-time control laws. The second strategy to solve the inverse geometric model is to use an algorithm based on an inverse kinematic model in a numerical integration process. In such an algorithm, the desired value of x and the initial condition of the joints are provided, and at the end of the iterative process a solution for that x is obtained.

## 3. Image Jacobian

In this section, we present the development of a linear camera model based on the pinhole camera model [28], and develop an image Jacobian matrix that relates the velocities of the end effector in three-dimensional space to the velocities in camera space, from the derivative with respect to time of the linear camera model.

We start from the pinhole camera model shown in Figure 3 where we can see point $P_{i}$ is projected onto the image plane through a segment passing via the origin of the camera coordinate frame (X,Y,Z). This way, we can relate one point in three-dimensional space to another in image space. To mathematically model such a projection, we first pose a homogeneous transformation from the global reference frame (x,y,z) to the camera reference frame (X,Y,Z) as follows:(25)$$\left[\begin{array}{c} X\\ Y\\ Z \end{array}\right]=\left[\begin{array}{cccc} R_{11} & R_{12} & R_{13} & X_{0}\\ R_{21} & R_{22} & R_{23} & Y_{0}\\ R_{31} & R_{32} & R_{33} & Z_{0} \end{array}\right]\left[\begin{array}{c} x\\ y\\ z\\ 1 \end{array}\right]=H\left[\begin{array}{c} x\\ y\\ z\\ 1 \end{array}\right]$$

Assuming that the camera is fixed, the matrix H is constant. With the hole camera model, it is known that the projection of a point represented in the frame (X,Y,Z) onto the frame $\left(x_{c},\;y_{c}\right)$ in the image plane is as follows:$$x_{c}=f\frac{X}{Z}\;y\;y_{c}=f\frac{Y}{Z}$$
where *f* is the focal length. The latter can be written in matrix form as:(26)$$\left[\begin{array}{c} x_{c}\\ y_{c}\\ 1 \end{array}\right]=\frac{1}{Z}\left[\begin{array}{ccc} f & 0 & 0\\ 0 & f & 0\\ 0 & 0 & 1 \end{array}\right]\left[\begin{array}{c} X\\ Y\\ Z \end{array}\right]=\frac{1}{Z}F\left[\begin{array}{c} X\\ Y\\ Z \end{array}\right]$$
where
$$F=\left[\begin{array}{ccc} f & 0 & 0\\ 0 & f & 0\\ 0 & 0 & 1 \end{array}\right]$$

Assuming that the focal length is constant, then *F* is constant.

The coordinates in the frame $\left(x_{c},\;y_{c}\right)$ are at the center of the image plane, and they are in length units. Another mapping is needed that moves the points to the frame typically used in digital imaging. That is, a (u,v) frame in the upper left corner of the image plane in which the direction of the *u* axis coincides with the direction of the $x_{c}$ axis while the direction of the *v* axis is in the opposite direction to that of the $y_{c}$ axis. In addition, we switch from longitudinal units to pixel units via the following homogeneous transformation:(27)$$\left[\begin{array}{c} u\\ v\\ 1 \end{array}\right]=\left[\begin{array}{ccc} m_{x} & 0 & p_{x}\\ 0 & -m_{y} & p_{y}\\ 0 & 0 & 1 \end{array}\right]\left[\begin{array}{c} x_{c}\\ y_{c}\\ 1 \end{array}\right]=K\left[\begin{array}{c} x_{c}\\ y_{c}\\ 1 \end{array}\right]$$
where $m_{x},\;m_{y}$ represents the number of pixels per unit distance, and $p_{x}=c_{x}m_{x}$ and $p_{y}=c_{y}m_{y}$ are the coordinates of the center of the image plane in pixel units. All these parameters are considered constant for a camera so that the matrix K is constant.

We can perform the composition of all these homogeneous transformations to establish a direct transformation between the frame (x,y,z) and the frame (u,v) in the following way:(28)$$\left[\begin{array}{c} u\\ v\\ 1 \end{array}\right]=\frac{1}{Z}\mathrm{KFH}\left[\begin{array}{c} x\\ y\\ z \end{array}\right]$$

Developing the multiplication of matrices, we obtain
(29)$$Z\left[\begin{array}{c} u\\ v\\ 1 \end{array}\right]=\left[\begin{array}{cccc} c_{11} & c_{12} & c_{13} & c_{14}\\ c_{21} & c_{22} & c_{23} & c_{24}\\ c_{31} & c_{32} & c_{33} & c_{34} \end{array}\right]\left[\begin{array}{c} x\\ y\\ z\\ 1 \end{array}\right]$$
where
$$c_{1j}=m_{x}\left(fR_{1j}+p_{x}R_{3j}\right)$$
$$c_{14}=m_{x}\left(fX_{0}+p_{x}Z_{0}\right)$$
$$c_{2j}=m_{y}\left(-fR_{1j}+p_{y}R_{3j}\right)$$
$$c_{24}=m_{y}\left(-fX_{0}+p_{y}Z_{0}\right)$$
$$c_{3j}=R_{3j}$$
$$c_{34}=Z_{0}$$
with j=1,2,3.

Since this mapping is composed of eleven independent parameters, we can use one of the terms of the resulting matrix as a scaling factor and divide the equation by that element. The element taken is the last element of the matrix, i.e., $Z_{0}$. With this, we obtain the following expression:(30)$$\frac{Z}{Z_{0}}\left[\begin{array}{c} u\\ v\\ 1 \end{array}\right]=\frac{1}{Z_{0}}\left[\begin{array}{cccc} c_{11} & c_{12} & c_{13} & c_{14}\\ c_{21} & c_{22} & c_{23} & c_{24}\\ c_{31} & c_{32} & c_{33} & c_{34} \end{array}\right]\left[\begin{array}{c} x\\ y\\ z\\ 1 \end{array}\right]$$

Which can be expressed as
(31)$$\rho \left[\begin{array}{c} u\\ v\\ 1 \end{array}\right]=P\left[\begin{array}{c} x\\ y\\ z\\ 1 \end{array}\right]$$
where
(32)$$P=\frac{1}{Z_{0}}\begin{array}{c} \left[\begin{array}{cccc} c_{11} & c_{12} & c_{13} & c_{14}\\ c_{21} & c_{22} & c_{23} & c_{24}\\ c_{31} & c_{32} & c_{33} & c_{34} \end{array}\right] \end{array}=\left[\begin{array}{cccc} P_{11} & P_{12} & P_{13} & P_{14}\\ P_{21} & P_{22} & P_{23} & P_{24}\\ P_{31} & P_{32} & P_{33} & 1 \end{array}\right]$$
and
(33)$$\rho =\begin{array}{c} \frac{Z}{Z_{0}} \end{array}$$
From the third row of (Equation 31), another definition of ρ can be extracted as follows:(34)$$\rho =P_{31}x+P_{32}y+P_{33}z+1$$

Note that P in (Equation 32) is constant and independent of the depth scale factor ρ, thus obtaining as a result the following relationship, which we call the linear camera model:(35)$$\rho \left[\begin{array}{c} u\\ v\\ 1 \end{array}\right]=\left[\begin{array}{cccc} P_{11} & P_{12} & P_{13} & P_{14}\\ P_{21} & P_{22} & P_{23} & P_{24}\\ P_{31} & P_{32} & P_{33} & 1 \end{array}\right]\left[\begin{array}{c} x\\ y\\ z\\ 1 \end{array}\right]$$

From this result, the image Jacobian can be developed. To accomplish this, we can recast Equation (Equation 35) starting with developing the product of ρ with the image coordinates obtaining
$$u=\frac{P_{11}x+P_{12}y+P_{13}z+P_{14}}{P_{31}x+P_{32}y+P_{33}z+1}$$
and
$$v=\frac{P_{21}x+P_{22}y+P_{23}z+P_{24}}{P_{31}x+P_{32}y+P_{33}z+1}$$
which, when restructured in matrix form, provides:(36)$$\left[\begin{array}{c} u-P_{14}\\ v-P_{24} \end{array}\right]=\left[\begin{array}{ccc} P_{11}-P_{31}u & P_{12}-P_{32}u & P_{13}-P_{33}u\\ P_{21}-P_{31}v & P_{22}-P_{32}v & P_{23}-P_{33}v \end{array}\right]\left[\begin{array}{c} x\\ y\\ z \end{array}\right]$$
or, in contracted form:(37)s(t)=P(s(t))x(t)
where
$$s\left(t\right)=\left[\begin{array}{c} u-P_{14}\\ v-P_{24} \end{array}\right]\;,$$
and
$$P\left(s\left(t\right)\right)=\left[\begin{array}{ccc} P_{11}-P_{31}u & P_{12}-P_{32}u & P_{13}-P_{33}u\\ P_{21}-P_{31}v & P_{22}-P_{32}v & P_{23}-P_{33}v \end{array}\right].$$

The vectors s(t) and x(t) represent the coordinates in image space (subtracting $\left(P_{14},P_{24}\right)$) and Cartesian space, respectively. Both vectors are functions of time since they represent the position of a target that can move in three-dimensional space at any time. Deriving with respect to time Equation (Equation 37), we obtain
(38)$$\dot{s}\left(t\right)=\dot{P}\left(s\left(t\right)\right)x\left(t\right)+P\left(s\left(t\right)\right)\dot{x}\left(t\right)$$
where
$$x\left(t\right)=\left[\begin{array}{c} x\\ y\\ z \end{array}\right]$$
$$\dot{s}\left(t\right)=\left[\begin{array}{c} \dot{u}\\ \dot{v} \end{array}\right]\;,$$
$$\dot{x}\left(t\right)=\left[\begin{array}{c} \dot{x}\\ \dot{y}\\ \dot{z} \end{array}\right]\;,$$
$$\dot{P}\left(s\left(t\right)\right)=\left[\begin{array}{ccc} -P_{31}\dot{u} & -P_{32}\dot{u} & -P_{33}\dot{u}\\ -P_{31}\dot{v} & -P_{32}\dot{v} & -P_{33}\dot{v} \end{array}\right].$$

The term $\dot{P}\left(s\left(t\right)\right)x\left(t\right)$ in (Equation 38) can be developed as
$$\dot{P}\left(s\left(t\right)\right)x\left(t\right)=\left[\begin{array}{c} \dot{u}\left(-P_{31}x-P_{32}y-P_{33}z\right)\\ \dot{v}\left(-P_{31}x-P_{32}y-P_{33}z\right) \end{array}\right]$$
$$\dot{P}\left(s\left(t\right)\right)x\left(t\right)=\left(-P_{31}x-P_{32}y-P_{33}z\right)\left[\begin{array}{c} \dot{u}\\ \dot{v} \end{array}\right]$$
and finally
(39)$$\dot{P}\left(s\left(t\right)\right)x\left(t\right)=\left(-\rho +1\right)\dot{s}\left(t\right).$$

Hence, we can restate (Equation 38) as follows:$$\dot{s}\left(t\right)=\left(-\rho +1\right)\dot{s}\left(t\right)+P\left(s\left(t\right)\right)\dot{x}\left(t\right)$$
$$\rho \dot{s}\left(t\right)=P\left(s\left(t\right)\right)\dot{x}\left(t\right)$$
$$\dot{s}\left(t\right)=\frac{1}{\rho }P\left(s\left(t\right)\right)\dot{x}\left(t\right)$$
(40)$$\dot{s}\left(t\right)=J_{csm}\left(s\left(t\right),\;x\left(t\right)\right)\dot{x}\left(t\right)$$
where
(41)$$J_{csm}\left(s\left(t\right),\;x\left(t\right)\right)=\frac{1}{\rho (x(t))}P\left(s\left(t\right)\right)$$

Thus, a direct relationship between the velocities in camera space and the velocities in three-dimensional space for each camera is obtained. This is of great value in the control context, since with this relation between velocities it is possible to define a closed-loop visual control laws using error signals defined as a function of the coordinates in image space. In order to establish an injective mapping in (Equation 40) a minimum of two cameras are required [6]. It can be observed that the image Jacobian $J_{csm}$ in (Equation 41) is composed of the scale factor ρ and the mapping P, which contains the vision parameters. In previous works [6,28,36], this ρ factor was assumed to be constant and equal to 1 (embedded in the vision parameters). While this approximation works locally, little is known about the size of the region of validity of such approximation. Moreover, the vision parameter needs to be calculated locally (refined) at each control iteration. In contrast, with the formulation presented here, the (constant) vision parameters need only to be determined once and the variable depth factor ρ can be obtained in several ways (for example by using the robot’s geometric model, when the robot is close to the target; or by using 3D reconstruction from visual information. which can be as simple as a linear approximation). This not only reduces the computational cost of the calculation but also, and more importantly, makes the model globally valid, which is paramount from the automatic control point of view.

## 4. Control Laws for the Reconfigurable Robot

Usually, the sought behavior of most closed loop controlled systems is of the form
(42)$$\dot{e}=-Ge$$
where e is the difference between a reference value and the current value of the system’s state vector and G is a gain matrix. If −G is Hurwitz, an exponential convergence of e to zero is achieved. This behavior can be achieved for the system at hand by designing the following control laws, following the coordinates chosen to express the system behavior. Error e can be defined as $e=q^{*}-q$, $e=x^{*}-x$ and $e=s^{*}-s$. That is, in joint space, operational (Cartesian) space or camera space, respectively.

On this basis, and considering the joint velocities as the control input, the following control law can be defined in joint space:(43)$$\dot{q}=Ge+{\dot{q}}^{*}$$
where $e=q^{*}-q$ and ${\dot{q}}^{*}$ is the reference velocity in joint space. It is common to have the information of the joint values by using encoders with which actuators are usually controlled. Therefore, it is natural to think of a control with an error signal in the joint space, but it is not common to know the desired joint value. This information is not always available and represents an additional problem, which can be addressed by using the inverse kinematic model, which in turn requires the knowledge of the reference position in Cartesian space, usually not straightforwardly available, and solving the redundancy problem beforehand. This problem becomes even more difficult when it comes to obtaining the reference velocity. Furthermore, these calculations are subjected to the influence of model errors that are a known source of estimation error for these elements. This constitutes an obstacle for the implementation of this control law.

On the same basis, for the dynamical system expressed in operational (Cartesian) space
(44)$$\dot{x}=J\dot{q}$$
the following control law can be defined using the Jacobian and the solution of the reconfiguration problem:(45)$$\dot{q}=J^{†}\left(Ge+{\dot{x}}^{*}\right)-\gamma \left(I-J^{†}J\right)\frac{\delta \mu }{\delta q}$$
where $e=x^{*}-x$ and ${\dot{x}}^{*}$ is the reference velocity in Cartesian space. In the case of a workspace error signal, it is common to calculate the end-effector position using the direct geometric model using the joint information. As for the target position, in general the desired position is also not known or requires some indirect measurement by using somewhat complex 3D reconstruction models, once again, sensible to errors in the models. Moreover, yet again, the tracking problem is difficult to carry out as they require real-time determination not only of the target position but of the target velocity. This constitutes an obstacle for the implementation of this control law.

Consider the dynamic system in camera space:(46)$$\dot{s}\left(t\right)=J_{csm}J\dot{q}$$
with the above conditions and using the mapping between the camera space and the operational space, the following control law can be defined:(47)$$\dot{q}=J^{†}J_{csm}^{†}\left(Ge+{\dot{s}}^{*}\right)-\gamma \left(I-J^{†}J\right)\frac{\delta \mu }{\delta q}$$
where $e=s^{*}-s$ and ${\dot{s}}^{*}$ is the reference (target) velocity in camera space. Notice that both the reference position and the end-effector position in camera space are directly measured with the vision sensor and thus no model is directly involved to acquire them. As for the reference velocity, it can be estimated from the measurements of the reference position using a range of algorithms reported in the literature (e.g., Kalman filter, etc.). For these reasons, camera-space control can be adapted to a wide variety of practical applications.

All of the above control laws yield the same closed loop controlled system (Equation 42). In practice, one limitation of the above control laws is the exponential decay itself, which produce high initial control signals and, as the robot approaches the target, it decreases exponentially, slowing down the convergence. This behavior can be improved by implementing a variation of the exponential decay control law, using increasing gains as follows. In the model, we define the error dynamics as a piece-wise dynamic system where each partition has defined exponential decay dynamics. This can be written as follows:(48)$$\dot{e}=\left\{\begin{array}{ccc} -G_{0}e & \mathrm{if} & |e|>d_{1}\\ -G_{1}e & \mathrm{if} & d_{1}>\left|e\right|>d_{2}\\ -G_{2}e & \mathrm{if} & d_{2}>\left|e\right|>d_{3}\\ \; & \vdots & \\ -G_{n}e & \mathrm{if} & |e|<d_{n} \end{array}\right.$$
where $G_{0}<G_{1}<\cdots <G_{n}$ are matrices of the form G=λI, where λ is any positive scalar, $d_{1}>d_{2}>...>d_{n}>0$ are the values of the error norm defining the partitions of the state space of the system.

In exponential decay control, as the value of e decreases, the value of $\dot{e}=-\mathrm{Ge}$ decreases in the same proportion, causing a high time of convergence to the target point. In the piece-wise control variation, we seek to propose the gains $G_{i}$ to increase in value as e decreases in value so that the rate of error dynamics approaches a linear decay rate, thus improving the speed of operation. If $\Delta d=|d_{i}-d_{i+1}|$ with i=1,2,…,n−1, we can define $G_{i}$ as
(49)$$G_{0}=\lambda \cdot I$$
and
(50)$$G_{i}=\prod_{1}^{i}\left(1+\frac{\Delta d}{d_{1}-\left(i-1\right)\Delta d}\right)\cdot G_{0}$$
Equation (Equation 50) guarantees that the eigenvalues of matrices $G_{i}$ increase exponentially, compensating for the decrease in the value of the error norm. Since all matrices $-G_{i}$ are Hurwitz in every partition whose union defines the entire state space, the dynamics of system (Equation 48) is asymptotically stable. This is a variation of the control law for all of the above control laws.

### 4.1. Encoders Based Control Laws

Consider the dynamical system (Equation 44) where $\dot{q}$ can be considered as the control input. Defining the closed-loop error dynamics (Equation 48) with $e=x^{*}-x$ and the solution to the redundancy problem (Equation 22), we can obtain the following control law:(51)$$\dot{q}=\left\{\begin{array}{ccc} J^{†}\left(G_{0}\left(e\right)+{\dot{x}}^{*}\right)-\gamma \left(I-J^{†}J\right)\frac{\delta \mu }{\delta q} & \mathrm{if} & |e|>d_{1}\\ J^{†}\left(G_{1}\left(e\right)+{\dot{x}}^{*}\right)-\gamma \left(I-J^{†}J\right)\frac{\delta \mu }{\delta q} & \mathrm{if} & d_{1}>\left|e\right|>d_{2}\\ J^{†}\left(G_{2}\left(e\right)+{\dot{x}}^{*}\right)-\gamma \left(I-J^{†}J\right)\frac{\delta \mu }{\delta q} & \mathrm{if} & d_{2}>\left|e\right|>d_{3}\\ \; & \vdots & \\ J^{†}\left(G_{n}\left(e\right)+{\dot{x}}^{*}\right)-\gamma \left(I-J^{†}J\right)\frac{\delta \mu }{\delta q} & \mathrm{if} & |e|<d_{n} \end{array}\right.$$

Another proposed control law is based on the closed-loop error dynamics (Equation 48) with the error defined in joint values: $e=q^{*}-q$ obtaining the control law:(52)$$\dot{q}=\left\{\begin{array}{ccc} G_{0}\left(e\right)+{\dot{q}}^{*} & \mathrm{if} & |e|>d_{1}\\ G_{1}\left(e\right)+{\dot{q}}^{*} & \mathrm{if} & d_{1}>\left|e\right|>d_{2}\\ G_{2}\left(e\right)+{\dot{q}}^{*} & \mathrm{if} & d_{2}>\left|e\right|>d_{3}\\ \; & \vdots & \\ G_{n}\left(e\right)+{\dot{q}}^{*} & \mathrm{if} & |e|<d_{n} \end{array}\right.$$

In this case, the redundancy problem is solved before the execution of the task to obtain the value of $q^{*}$ using one of the algorithms presented in the previous section.

### 4.2. Control Law in Vision Sensor Space

Following the guidelines presented for developing control laws based on encoder signals and taking the dynamic system (Equation 46) where $\dot{q}$ is the control input. The resulting control law is:(53)$$\dot{q}=\left\{\begin{array}{ccc} J^{†}J_{\mathrm{csm}}^{†}\left(G_{0}\left(e\right)+{\dot{s}}^{*}\right)-\gamma \left(I-J^{†}J\right)\frac{\delta \mu }{\delta q} & \mathrm{if} & |e|>d_{1}\\ J^{†}J_{\mathrm{csm}}^{†}\left(G_{1}\left(e\right)+{\dot{s}}^{*}\right)-\gamma \left(I-J^{†}J\right)\frac{\delta \mu }{\delta q} & \mathrm{if} & d_{1}>\left|e\right|>d_{2}\\ J^{†}J_{\mathrm{csm}}^{†}\left(G_{2}\left(e\right)+{\dot{s}}^{*}\right)-\gamma \left(I-J^{†}J\right)\frac{\delta \mu }{\delta q} & \mathrm{if} & d_{2}>\left|e\right|>d_{3}\\ \; & \vdots & \\ J^{†}J_{\mathrm{csm}}^{†}\left(G_{n}\left(e\right)+{\dot{s}}^{*}\right)-\gamma \left(I-J^{†}J\right)\frac{\delta \mu }{\delta q} & \mathrm{if} & |e|<d_{n} \end{array}\right.$$
where ${\dot{s}}^{*}$ is estimated using the numerical derivation of $s^{*}$ from the measurements in the previous control cycles.

## 5. Experiments

This section presents the experimental setup implemented to validate the control laws designed in the previous sections. First, a brief description of the hardware and software will be given of the different parts of the test bench used.

### 5.1. Hardware

We used a reconfigurable variant of the delta PARALLIX LKF-2040 robot. The kinematic description of the robot is presented in Section 2. The robot is composed of four maxon 380961 motors, each of them controlled with a PICSERVO SC board. Communication with the control boards is serial over USB. The boards control the actuators using an internal PID controller and receive speeds as commands. A conveyor has been placed inside the workspace of the robot to allow for the tracking of moving objects.

The vision system is composed of two Allied Vision cameras of the Alvium line at 1296 × 964 resolution capturing at 130 frames per second through UBS 3.0. The cameras are located at a distance of approximately 1.5 m from the robot. Each camera is 1 m apart. A personal computer with Intel(R) Core(TM) i7-8700 CPU @ 3.20GHz of 6 and 12 cores, 16 gigabytes of RAM, was used to control the robot controller boards, the vision system, and to perform the control law calculation. Figure 4 shows a picture of the full system and Figure 5 shows an schematic of the hardware system and its workflow.

### 5.2. Software

The control system was programmed using Python 3.10. Numerical computations, necessary in the control law, were performed using NumPy 1.24.3 [37]. Image capture was handled via VimbaPython 6.0 SDK provided by the camera manufacturer while image processing used OpenCV 4.7 [38]. Finally, the PyQt 5.15.9 [39] library was used to generate a graphical interface and thread management. The program is composed of three main threads:(i)A main thread for all the cameras,(ii)A thread to send commands to the robot,(iii)A thread for the control law calculation.

The control cycle time obtained with the aforementioned set-up was 16 ms.

### 5.3. Testing Configuration

First, static positioning tasks are used to validate and compare the performance of exponential convergence control laws ((Equation 45), (Equation 43) and (Equation 47)) versus the piece-wise variants ((Equation 51), (Equation 52) and (Equation 53)) with reference velocity equal zero. Reference positions were distributed in the robot workspace and the robot was commanded to reach those positions using the designed control laws. For the vision-based control law, the vision parameters were initially estimated using what has been referred to as a pre-plan. During this procedure, the robot is positioned at a series of points covering its workspace. For these tests, 410 measurements were used, evenly distributed throughout the robot’s workspace, with 50 mm spacing on the *X* and *Y* axes and 25 mm spacing on the *Z* axis.

Second, in order to fully validate the vision-based control law (Equation 53) and evaluate its performance, two types of experiments are considered:(i)The first set of experiments for the CSM control law consists of static positioning tasks. Four position tasks are performed three times each. The positions are chosen randomly inside the workspace of the robot. The maneuver is considered successful when an error vector norm of less than 2 pixel is achieved during, at least, 40 control cycles. This error vector contains the coordinates in image space of the two cameras. Speed compensation is set to zero to avoid noise input due to the velocity estimator.(ii)The second set of experiments for the CSM control law consists of tracking an object moving, on the conveyor, following a linear trajectory at constant speed. No a priori information about the target trajectory is used in the experiments. The target is positioned on a conveyor belt placed within the robot’s workspace. The robot first moves to the target object. The conveyor is started only once a static position has been achieved using the previously mentioned conditions. The conveyor moves at a constant speed with the robot following the object’s position. The test is stopped when the object reaches the edge of the robot’s workspace. The control law with speed compensation is used.

Figure 6 shows the general operation of the control system using the example of visual control for positioning tasks, where the stop criterion corresponds to when the error norm is less than 2 pixels. The operation is similar for the tracking task, with the difference that the stop criterion corresponds to the target leaving the assigned tasking area.

## 6. Results and Discussion

### 6.1. Results

#### 6.1.1. Exponential Convergence Control Laws vs. Piece-Wise Variants

Static positioning tasks using exponential convergence control laws ((Equation 43), (Equation 45) and (Equation 47)) were executed to compare their performance with the corresponding piece-wise variants ((Equation 51)–(Equation 53)). Only results for the control laws in the operational space are shown, although similar results were obtained for the control laws in the other spaces. Figure 7 and Figure 8 show the error norm and control signal, respectively, for control law (Equation 45). Similarly, Figure 9 and Figure 10 show the error norm and control signals for the piece-wise variant (Equation 51). Comparing the images, which consist of the same positioning task, it can be seen how the piece-wise variant is effective in improving the convergence time from 4 s to 1.8 s, while preserving the steady state zero error of the exponential convergence control law.

#### 6.1.2. Control Law in Vision-Sensor Space

For the first set of experiments (i.e., static positioning) for the vision-based control law, an average error of 0.604 mm was obtained. Table 1 shows more detail about these results. The maximum and minimum errors coincide with the target point farthest from the cameras and the target point closest to the cameras. This can happen because the pixel/mm ratio decreases as the distance increases (this phenomenon is also observed in the tracking tasks reported below).

Figure 11 shows the position error norm in image coordinates, measured in pixels. It can be seen how the behavior of the norm is similar to a linear decay in most of the robot’s path. This is a direct effect of the piece-wise control law. Figure 12 shows the velocity commands sent to the control boards. The angular velocities (${\dot{\theta }}_{1i}$) of the actuated rotational joints are measured on the scale to the left, while the velocity of the prismatic reconfiguration joint ($\dot{R}$) is measured on the scale to the right. In all cases, the signals tend to zero.

For the tracking task of the CSM control law, for an object moving at constant speed, average drag errors were measured between 2.5 mm and 11.5 mm for speeds ranging from 3.5 cm/s to 12.7 cm/s. The tracking error increases as the belt speed increases. The results are shown in the Table 2. Figure 13, Figure 14 and Figure 15 show the tracking error in both the image and operational spaces, for the experiments at different conveyor belt speeds. The tracking error in the image space is represented by the blue line and is measured on the left hand scale while the tracking error in the operational space is represented by the violet line and is presented on the right hand scale. At the beginning of the experiments at 3.5 cm/s and 9.3 cm/s, the object is motionless and the robot positions on the target; then, the object starts moving at a constant speed and the robot tracks the target. For the experiment at 12.7 cm/s, the target starts moving from the beginning of the experiment. Table 2 shows the tracking errors in operational space. The tracking error consists of the average of the error norm from the time the target starts its movement to the end of the experiment when the target leaves the working area. This time can vary depending on the tracking speed between 2.5 s and 5 s. The working area allows for a trajectory of the object covering a distance of 38 cm. We can see that the average error is always a small fraction of the distance covered by the target and stays always below 3%.

### 6.2. Discussion

In the comparison between exponential convergence control laws and piece-wise variants, it can be seen how the piece-wise variant is effective in improving the convergence time to about half the original time, while preserving the steady state zero error of the exponential convergence control law. In practical applications, this represents a significant advantage, as it can accelerate processes.

With respect to the control laws designed in the vision sensor-space, in static positioning tasks, an average positioning error of 0.6 mm was obtained. This implies an improvement in positioning precision over previous implementations of LCM-CSM/CSM control laws [6,28,36] where an accuracy of about 1 mm was achieved. This improvement can be attributed both to the implementation of the LCM-CSM Jacobian matrix that includes the depth scale information and to the higher resolution of the cameras. In the tests performed, a loss of accuracy in the positioning task was observed as the target moves away from the cameras. This can be explained by the fact that the farther away the cameras, the lower the pixel/mm correspondence.

For the constant velocity trajectory tracking task, tracking errors of 2.5 mm to 5 mm were achieved for velocities of 6.5 to 12.7 cm/s. These results are competitive with other results reported in the literature. Similar to the positioning task, there is an increase in error as the target moves away from the cameras. It was found that the velocities estimated by averaging the last 10 numerical derivatives produce very noisy signals that are transferred to the control signal. This can affect the performance of the robot and generate a higher tracking error compared to a cleaner velocity signal.If you have a polynomial estimate of the trajectory in the last 10 steps, the velocity can be calculated analytically.

## 7. Conclusions

In this article, the Jacobian analysis and the design of vision-based control laws for a reconfigurable, redundant delta-type parallel robot are presented. We investigated a target-tracking task where the target trajectory is not known in advance. Also, the image Jacobian ($J_{csm}$) based on the LCM-CSM approach is revisited and the vision parameters are shown to be constant and independent of a depth scale factor, the function of the depth of the observed features, which is variable. This allows us to very easily update the image Jacobian matrix at each control cycle with a very low computational cost.

Using the proposed control strategy, in the case of static objectives, the average positioning error was 0.6 mm with a standard deviation of 0.445 mm. These results are competitive when compared against a number of industrial applications. Regarding mobile object tracking tasks, the results show a tracking errors of 2.5 mm, 3.9 mm and 5 mm for targets moving along a linear trajectory at speeds of 6.5, 9.3 and 12.7 cm/s, respectively, with a maximum final static positioning errors of 1.53 mm once the object stopped.

Parallel robots require specialized control designs because of their model complexity and open problems, which prevent simply extending existing robot manipulator controls. Moreover, while a reconfigurable parallel manipulator increases the versatility of the manipulator and tackles some of its limitations, it also increases the complexity of the robot and requires an even more specialized control. Delta robots are very popular in industry, particularly in “pick and place” tasks; however, these applications require on-line adjustments, specification changes and the tracking of moving targets. Under these circumstances, vision-based control provides the necessary flexibility over traditional control approaches and, in the authors’ opinion, the proposed vision-based control via CSM provides a satisfactory solution for said tasks.

### Future Work

The results presented in this article are a starting point for the topics addressed here. On the one hand, it is possible to deepen the analysis of the characteristics resulting from the reconfiguration of the robot. On the other hand, the presented piecewise control law shows a significant improvement in the convergence rate compared to the exponential decay control law. Comparisons with other control laws, such as PID, which are traditionally one of the first choices for controlling a system, can shed light on the full potential of the proposed law. The development of a control methodology around the proposed law is desirable. Considering the VLCM-CSM visual control, comparisons can be made between the Jacobian in this article with those reported in other papers to obtain a better sense of its effectiveness.

## Figures and Tables

**Figure 1 sensors-23-07039-f001:**
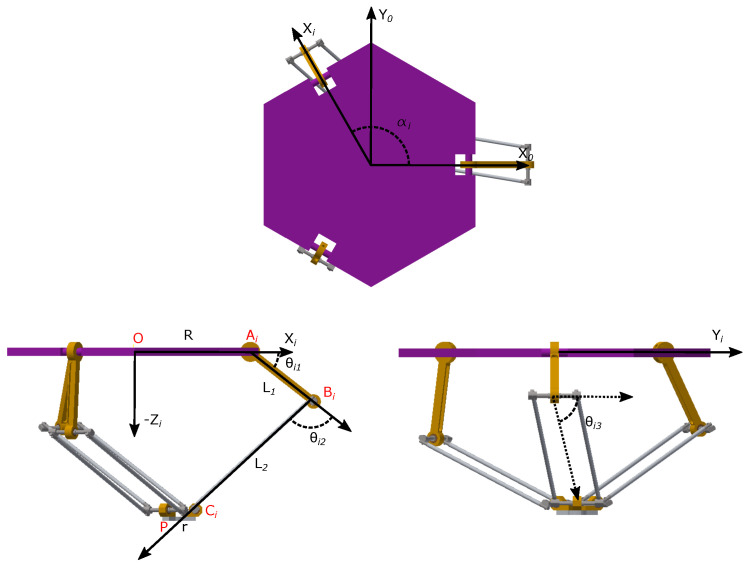
Kinematic chains of the delta robot.

**Figure 2 sensors-23-07039-f002:**
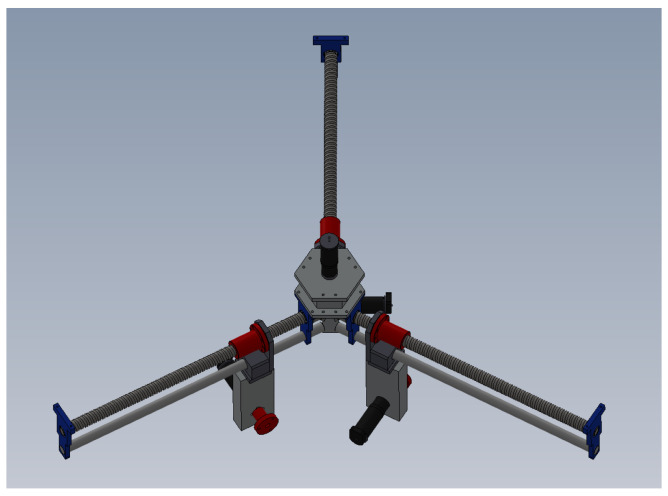
Reconfiguration mechanism for the delta robot.

**Figure 3 sensors-23-07039-f003:**
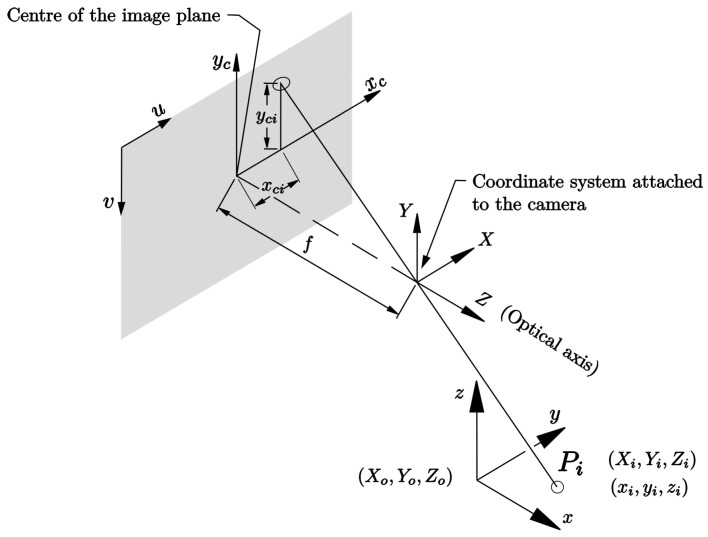
Pinhole camera model.

**Figure 4 sensors-23-07039-f004:**
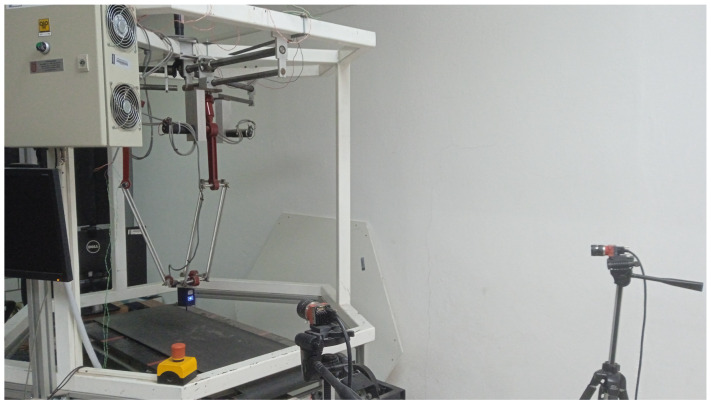
The reconfigurable delta robot and camera system.

**Figure 5 sensors-23-07039-f005:**
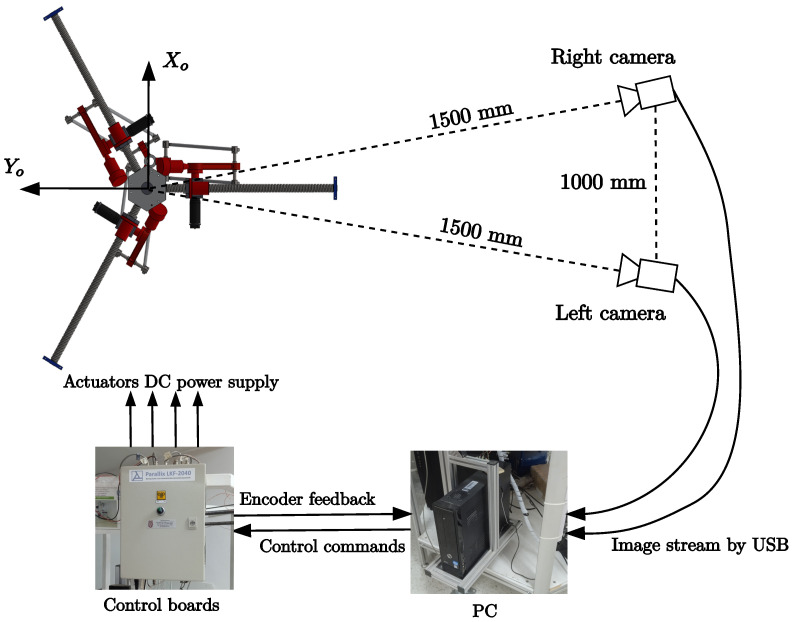
Schematic of the hardware system and its workflow.

**Figure 6 sensors-23-07039-f006:**
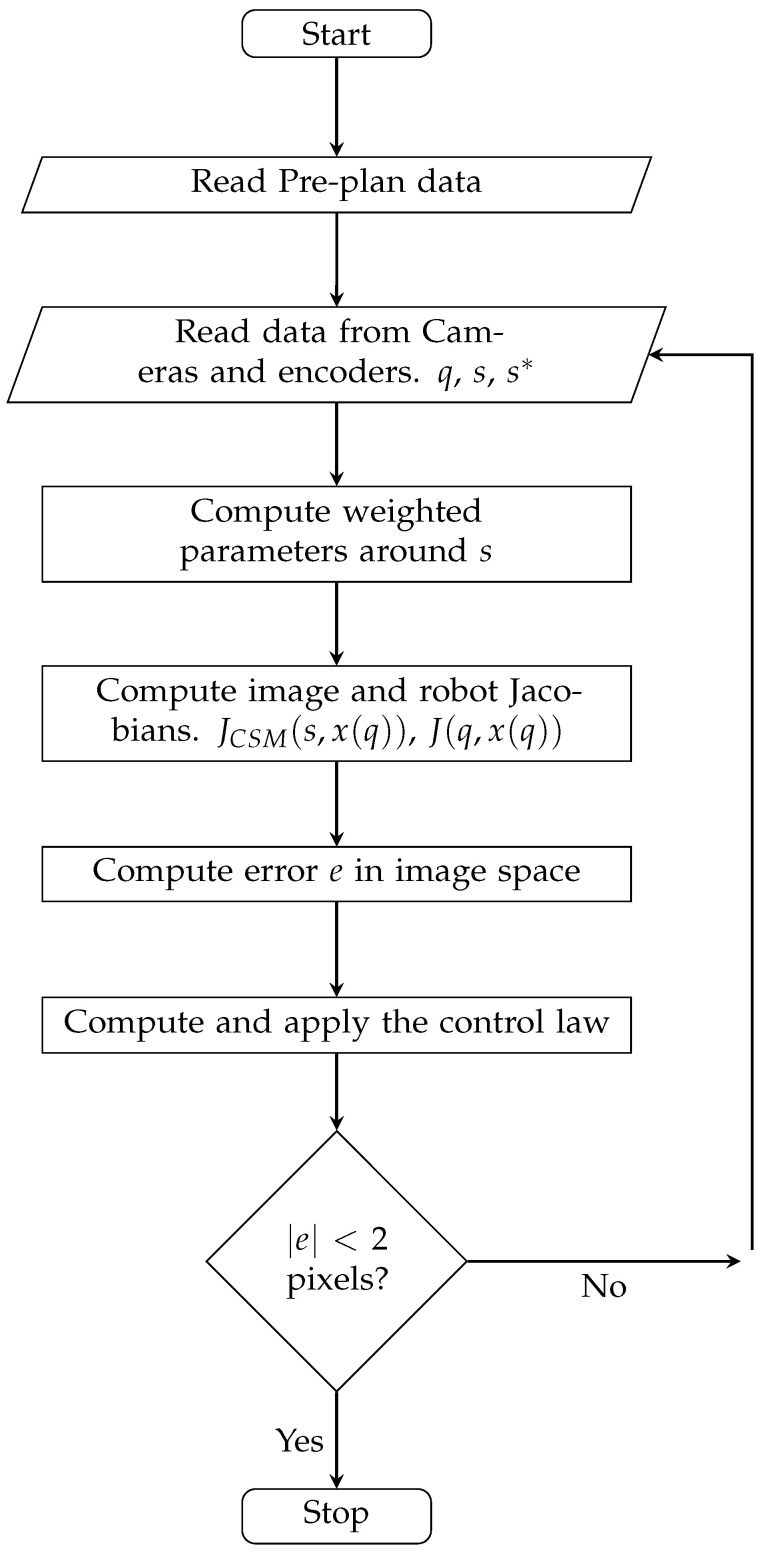
General operation of the control system.

**Figure 7 sensors-23-07039-f007:**
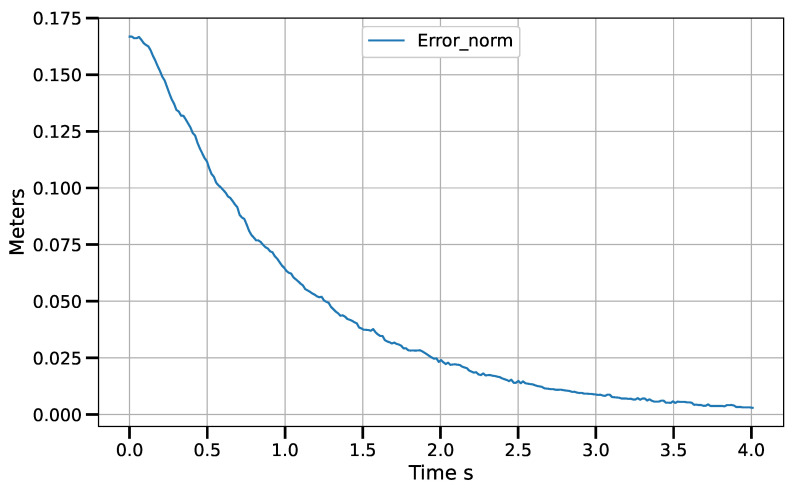
Error norm vs. time for control law $\dot{q}=J^{†}G\left(e\right)-\gamma \left(I-J^{†}J\right)\frac{\delta \mu }{\delta q}$ with $e=x^{*}-x$.

**Figure 8 sensors-23-07039-f008:**
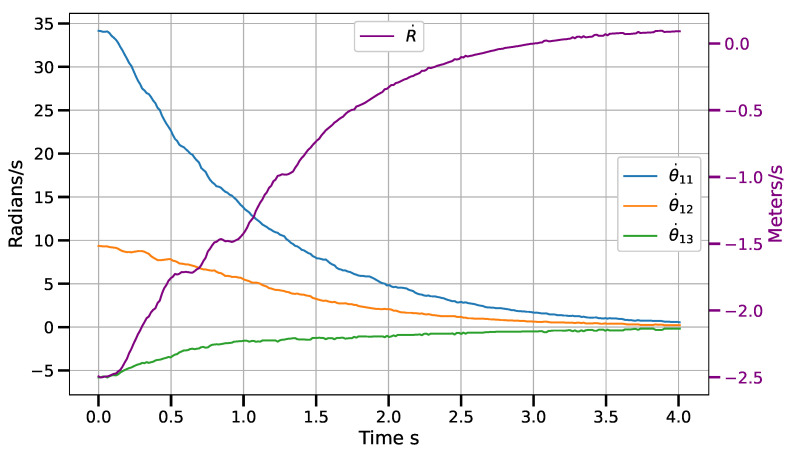
Angular joint velocity (radians/s) vs. time for control law $\dot{q}=J^{†}G\left(e\right)-\gamma \left(I-J^{†}J\right)\frac{\delta \mu }{\delta q}$ with $e=x^{*}-x$.

**Figure 9 sensors-23-07039-f009:**
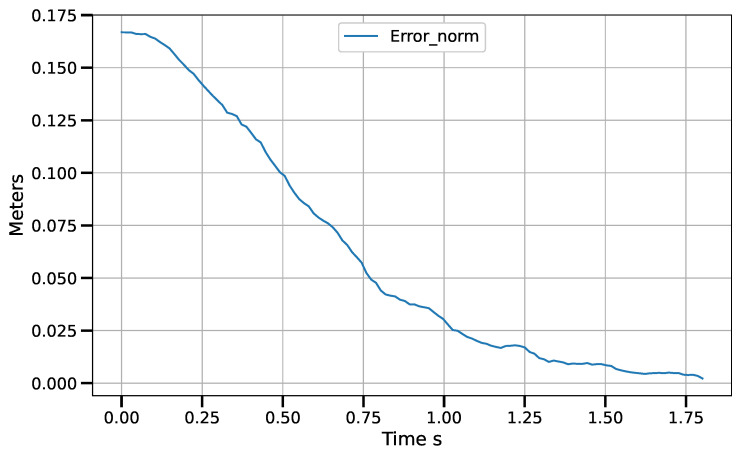
Error norm vs. time for control law (Equation 51) with $e=x^{*}-x$.

**Figure 10 sensors-23-07039-f010:**
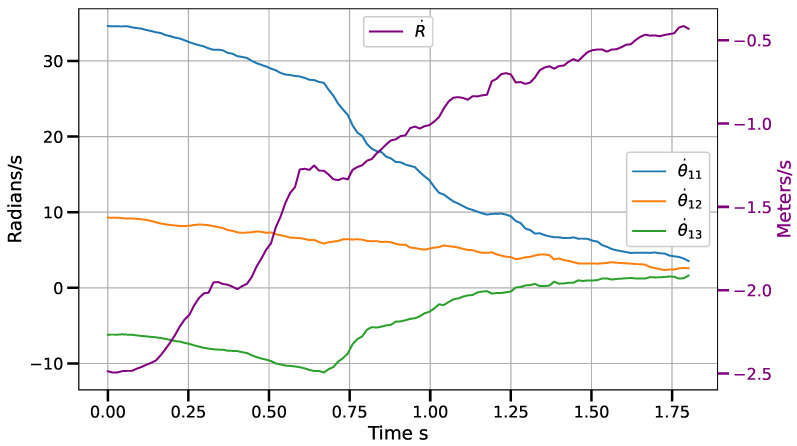
Angular joint velocity (radians/s) vs. time for control law (Equation 51) with $e=x^{*}-x$.

**Figure 11 sensors-23-07039-f011:**
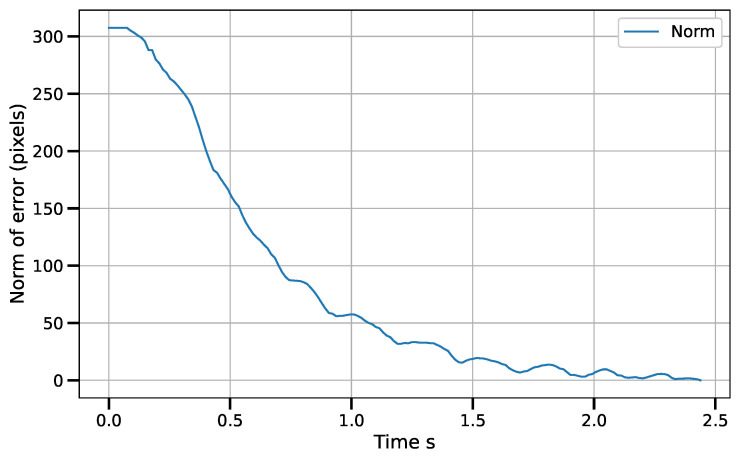
Positioning error norm in image space.

**Figure 12 sensors-23-07039-f012:**
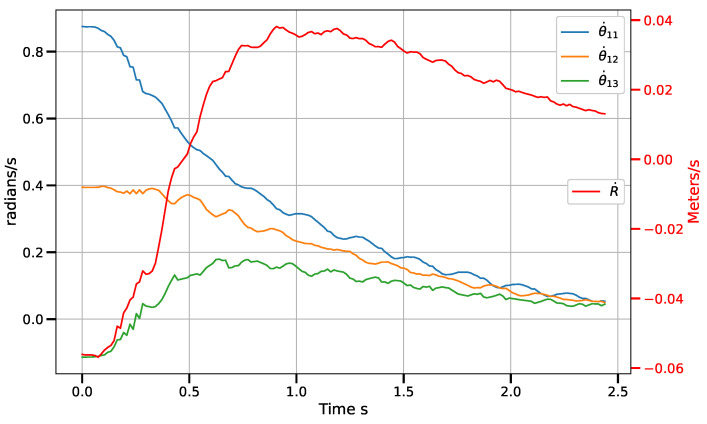
Velocity control signal.

**Figure 13 sensors-23-07039-f013:**
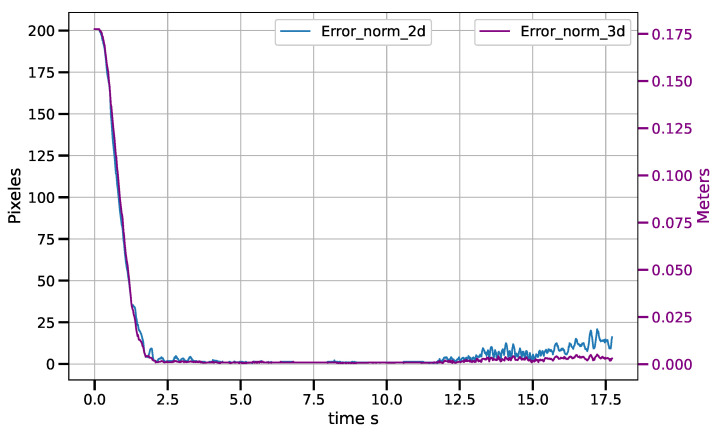
Tracking error in image and operational spaces for a speed of 6.5 cm/s.

**Figure 14 sensors-23-07039-f014:**
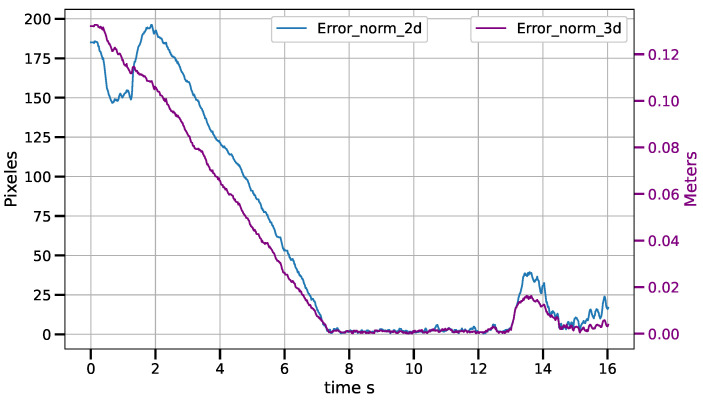
Tracking error in image and operational spaces for a speed of 9.3 cm/s.

**Figure 15 sensors-23-07039-f015:**
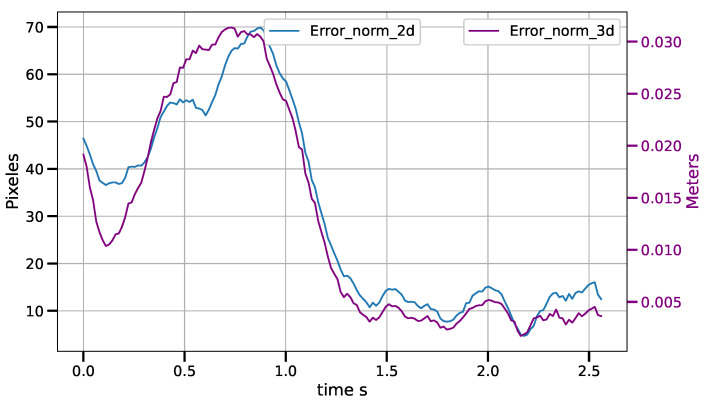
Tracking error in image and operational spaces for a speed of 12.7 cm/s.

**Table 1 sensors-23-07039-t001:** Position error for positioning task.

Error (mm)	
Average	0.600
Max	1.530
Min	0.172
Std. Dev.	0.445

**Table 2 sensors-23-07039-t002:** Tracking error in operational space for constant velocity task.

Conveyor Speed	Average Error
6.5 cm/s	2.5 mm
9.3 cm/s	3.9 mm
12.7 cm/s	11.5 mm

## Data Availability

Data available upon reasonable request.

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
