# Peer review of "Depth-Dependent Control in Vision-Sensor Space for Reconfigurable Parallel Manipulators"

_sensors, 2023, doi:10.3390/s23167039_

Round 1
Reviewer 1 Report
In this paper, the kinematic model of a reconfigurable delta robot is obtained by modeling it as a redundant robot. Control laws in the vision-sensor space and in the 3D space are then designed and implemented to control this robot in target tracking tasks.
The article covers interesting and scientifically relevant topics, the mathematical treatment is rigorous and satisfactory, however, a minor revision is needed to improve the quality of the work and give more emphasis to the novelty and strengths of the research. In some parts of the article, the correlation of covered topics is not clear.
In detail:
- The abstract is rather long and also not very clear about the work done and its novelty and strength. It is not persuasive to the reader.
- In the introduction few references are given about the state of the art in recent years, it is necessary to contextualize the problem better and highlight the points still open and how this article addresses them.
- In equation 7 and 8 the bold used for B is wrong.
-Line 136 grammatical error (Do do so we multiply…).
- The correlation between equation 32 and 33 is not clear, and it is also not clear in line 188 and following lines how the depth-dependent control is handled and its strength and usefulness for the solved problem (it is stated in the title so it is a crucial point but not well explained in depth in the text).
-Providing more insight into the camera calibration and the communication/control protocol used to control the robot, perhaps even a video or photos would be helpful to give strength to the work and more clarification to the reader.
-Too many references form the author and co-authors.
- It is not clear how the redundancy and the reconfiguration via R parameter helps and improves the system performance on the tracking task
Moderate editing of English language is required
Reviewer 2 Report
In this paper, the kinematic model of a reconfigurable Delta robot is obtained by modeling it as a redundant robot. Control laws in the vision-sensor space and in the 3D space are then designed and implemented to control this robot in target tracking tasks. No a priori information about the target trajectory is required. The contribution of this paper is not obvious. There are several flaws existing in this paper as following.
1. The abstract is vague; the background, method, and conclusion should be reflected in the abstract. Please revise the abstract carefully.
2. References cited in the frontier section should describe the academic contributions and the links with the viewpoints of the paper, such as line 42 [7-10], line 55 [13-16], and line 68 [18-20].
3. There is an error in Eq.(5), where R is a structural parameter, not a variable.
4. The dimension of formula Eqs(18-20) are wrong, please modify it.
5. How does image Jacobian (Section 3) relate to the coordinates of parallel robots? What is the conversion relationship between the two? Please give a clear relationship between the parallel robot's joint space, operation space, and visual space.
6. What is the control method adopted in the paper? In section 4, how can the proposed control system prove stable?
7. In Fig.3, what are the criteria for judging error convergence?
8. Please explain in detail the meaning of the data in Table 2. The reviewers do not consider these data sufficient to illustrate the accuracy of error tracking.
Reviewer 3 Report
The authors present a vision-based control for Delta robotic structure, using control- laws to command the robot in target tracking tasks.
The topic of the paper is interesting and could provide valuable information for readers, but there are some aspects that should be taken into consideration, in order to increase the value of the paper and to highlight the work of the authors.
The first aspect that should be addressed is related to the abstract, which in its current form provides too much information that belongs to the introduction section. The abstract is an overview of the paper and should contain a compressed part where is stated the topic, another part where is explained the issues that the paper address, the methods used to resolve these issues, the results and the conclusion.
To respect this structure, I kindly recommend that authors compress the information presented in the first part of the abstract and state the purpose of the paper earlier. I also suggest introducing the results and a brief conclusion.
Another important part of the paper that authors should revise is the introduction section considering the followings aspects:
- In its current form, the introduction section lacks substantial information regarding the state of the art for vision control based on parallel robots. I recommend that the authors expand this section to provide a comprehensive overview of the existing research and developments in this field.
- I observed that the aim of the paper is not clearly presented in the introduction section. It is recommended that the authors explicitly state the objective or purpose of their research within this section to provide readers with a clear understanding of the paper's focus and direction.
- Additionally, I recommend the introduction of a short description for the following sections of the paper (at the end of this section).
Other comments are related to the description of the forward geometric model, which in my humble opinion is too detailed as this structure is well known and completely solved in multiple papers. In the forward kinematic model, please explain how you are creating the redundancy (time variation) of R. If you are using changes in geometry, please add a schematic drawing to better explain the modifications of R.
I would also suggest the introduction of a schematic representation of the hardware system and its workflow described on page 14.
Considering the importance of references in a scientific paper, would be very beneficial for readers to have the entire list of references. It has come to my attention that while the text references up to number 52, the references section only includes citations up to number 38. I suggest expanding the references section to include all the cited sources in the text, ensuring that it aligns with the references mentioned throughout the paper.
Best regards!
None
Reviewer 4 Report
Dear authors,
Please consider the following suggestions of improvement:
-A paragraph with the paper structure should be included at the end of introduction;
-Tables captions should be included in the bottom;
-Table 1 error precision should be the same for all values (0.600; 1.530;…);
-A paragraph at the end of conclusion with some future work suggestions could be included.
Good use of English. Minor issues required.
Round 2
Reviewer 2 Report
All my comments have been properly accommodated.
Reviewer 3 Report
All my suggestions have been implemented within the paper.
Best regards!